# Benzimidazole–Pyrimidine Hybrids: Synthesis and Medicinal Properties

**DOI:** 10.3390/ph18081225

**Published:** 2025-08-19

**Authors:** Maria Marinescu, Christina Zalaru

**Affiliations:** Department of Inorganic Chemistry, Organic Chemistry, Biochemistry and Catalysis, Faculty of Chemistry, University of Bucharest, Soseaua Panduri, 030018 Bucharest, Romania

**Keywords:** hybrids, pyrimidine, benzimidazole, therapeutics, anticancer, antimicrobial, antiulcer, antiviral, anti-inflammatory, anti-Alzheimer’s, antioxidant

## Abstract

**Background**: Heterocyclic compounds represent a key class of compounds in medicinal chemistry. Both benzimidazoles and pyrimidines are essential heterocycles in medicinal chemistry, with various therapeutic properties. Recent literature presents a series of hybrid heterocyclic compounds, as their medicinal properties are generally improved compared to those of single heterocyclic rings. **Methods**: A literature search was conducted across relevant scientific literature from peer-reviewed sources, using keywords, including “benzimidazole”, “pyrimidine”, “Biginelli”, “benzimidazole-pyrimidine hybrids”, “anticancer”, “antiviral”, “antimicrobial”, and “anti-inflammatory”. **Results**: In this review, benzimidazole–pyrimidine hybrids are reported as anticancer, antimicrobial, antiviral, anti-inflammatory, analgesic, antiulcer, antidepressant, anti-Alzheimer’s, or antioxidant agents, with activities even better than those of existing drugs. The IC_50_ values for these anticancer hybrids are in the nanomolar range, which signifies potent anticancer agents. It can be mentioned here that the anticancer hybrid Abemaciclib, as a CDK4/6 inhibitor for the treatment of certain types of breast cancer, was approved in 2017. The antimicrobial activity of these hybrids proved especially potent against a broad variety of infections, with MIC values in the range of µM or even nM. Moreover, these hybrids exhibited good antiviral properties against SARS-CoV-2, HIV-1, and the hepatitis C virus. The hybrids also functioned as JAK3 inhibitors, COX-1 inhibitors, and MAO-A inhibitors. **Conclusions**: This review presents synthesis methods of benzimidazole–pyrimidine hybrids, their medicinal properties, and SAR studies reported in the last 20 years. For almost every therapeutic activity, SAR studies have revealed the essential presence of a substituent on the aromatic rings or between the two benzimidazole and pyrimidine nuclei.

## 1. Introduction

Since the medicinal properties of hybrid compounds are typically better than those of simple heterocyclic compounds, which only contain one type of heterocycle in the molecule, organic compounds with two or more distinct heterocyclic rings are being reported more frequently in current medicinal chemistry as possible therapeutic compounds [1,2,3,4,5]. Thus, hybrids created by combining two well-known pharmacophores have novel, peculiar characteristics, with fantastic properties that are determined by the unique structure of the newly created molecule [6]. This new design is highlighted in the efficacy of the new drug defined by the physicochemical parameters, namely absorption, distribution, the mechanism of interaction between the hybrid and its cellular target, metabolism, excretion, and toxicity [7,8,9]. Recent articles and reviews report a series of hybrid compounds with diverse therapeutic properties, such as coumarin–triazole [10], metronidazole–berberine [11], quinoline–triazole [12], benzimidazole–triazoles [13], benzimidazole–pyrazole [14,15,16], benzimidazole–morpholine [17], and of course many others.

Benzimidazole, or 1*H*-benzimidazole, a bicyclic heterocyclic aromatic compound in which a benzene ring is fused to the “4” and “5” positions of an imidazole ring, is a crucial part of vitamin B_12_, and only the imidazole ring scaffold is present in several natural compounds, such as histidine and purines. Due to its presence in many drugs, benzimidazole is also a key component in medicinal chemistry [17]. Recent articles indicate various routes for the synthesis of benzimidazoles, such as the coupling of 1,2-diaminobenzenes with carboxylic acids (Phillips–Ladenburg reaction), the coupling of 1,2-diaminobenzenes with aldehydes and ketones (Weidenhagen reaction), or the rearrangement of quinoxalinones [14,16].

The pyrimidine ring is present in vitamin B_1_ (thiamine) and several natural compounds, such as the pyrimidine bases uracil, thymine, and cytosine, which are fundamental building blocks for DNA and RNA synthesis and are essential for cellular functions [18]. Various methods for the synthesis of pyrimidines are studied in the literature, such as two-component cycloadditions, like the [5 + 1] annulation of enamidines, [4 + 2] cycloadditions, [3 + 3] cycloadditions, three-component cycloadditions, or the Biginelli reaction [18,19].

In this review, starting from the fact that both the pyrimidine [18,19,20] and the benzimidazole [21,22,23] nuclei are found as key nuclei in drugs with various therapeutic applications, such as anticancer [24,25], antibacterial [26,27], antifungal [28,29], antiviral [30,31], antidiabetic [32,33,34], antiulcer [35,36], antioxidant [37,38], anti-Alzheimer’s [39,40], antidepressant [41,42], anti-Parkinson’s [43,44], anticonvulsant [45,46], and anti-inflammatory [47,48], we aimed to review the synthetic methods of pyrimidine–benzimidazole hybrids, the medicinal properties of aforementioned hybrids, as well as the reported structure–property relationships (SAR). When required, a number of examples from the literature were provided for compounds with superior biological activity, along with a description of the several studies that were conducted on them (in vitro, in vivo, in silico).

To the best of our knowledge, this is the first review to examine the synthesis and therapeutic properties of pyrimidine–benzimidazole hybrids.

This review’s database search strategy involved using keywords that appear in the title, such as “pyrimidine”, “benzimidazole”, “benzimidazole-pyrimidine hybrids”, “Biginelli reaction”, “anticancer”, “anti-inflammatory”, “antimicrobial”, “antiviral”, and so forth, or therapeutic properties, across a variety of websites, including ACS Publications, PubMed, MDPI, Science Direct, Springer, The Royal Society Chemistry, and Taylor & Francis. Articles from the past ten years have often been chosen.

## 2. Anticancer Benzimidazole–Pyrimidine Hybrids

Cancer remains one of the most relentless contemporary diseases, increasingly common in people of all ages, social statuses, and lifestyles [49]. Finding new anticancer compounds remains a target of medicinal chemistry, for which nitrogen heterocyclic compounds constitute an advantageous choice, proven by recent studies [50].

Benzimidazole–pyrimidine compounds represent a good choice in the treatment of cancer, as proven by recent studies conducted on the hybrids discussed in this article. Abemaciclib (Figure 1), the newest benzimidazole–pyrimidine hybrid CDK4/6 inhibitor for the treatment of certain types of breast cancer, approved in 2017, is the drug that demonstrates the efficacy of the presence of the two heterocyclic rings in its structure [51,52,53,54]. Goetz et al. (2024) demonstrated in analyses from the monarchE study that protocol-mandated dose reductions from 150 mg to 100 mg or 50 mg for patients with node-positive, hormone receptor-positive, human epidermal growth factor 2-negative, high-risk early breast cancer did not impair the effectiveness of adjuvant abemaciclib [55]. In another study, monarchE has demonstrated that abemaciclib added to standard adjuvant ET aromatase inhibitors [AIs] and/or antiestrogens with or without ovarian suppression (ET), significantly improves Invasive Disease-Free Survival (IDFS) in women and men with hormone receptor-positive HR+, hormone receptor-negative HER2, and node-positive early breast cancer (EBC), at high risk of early recurrence [56,57,58,59]. Kalinsky et al. (2024) found Abemaciclib plus Fulvestrant significantly improved PFS (Progression-Free Survival) after disease progression on previous CDK4/6i (cyclin-dependent kinase 4/6 inhibitors) + ET in patients with HR+, HER2– ABC, providing these patients with an extra alternative for targeted treatment [60]. Shao et al. (2014) reported the synthesis of anticancer hybrids **3**–**6**, from 2-(chloromethyl)-1*H*-benzo[*d*]imidazole **1** and 2-mercapto-6- phenylpyrimidin-4-ol **2** in three steps (Figure 1) [61]. Higher yields are observed for compounds **6a**–**6e** compared to compounds **5a**–**5k** due to the presence of the chlorine atom in the “5” position of the benzimidazole nucleus. All compounds had good anticancer activity when tested in vitro on four human cancer cell lines, including MCF-7 (human breast cancer cell line), MGC-803 (human gastric cancer cell line), EC-9706 (human esophageal cancer cell line), and SMMC-7721 (human liver cancer cell line) using the MTT assay method, as can be seen from the IC_50_ (concentration required to achieve 50% inhibition of the tumor growth) values in Figure 1 for the hybrids and the standard 5-fluorouracil (5-FU). The most active compounds were **5a**–**5b** and **6a**–**6b**, with IC_50_ values of 1.33–20.50 µM and 1.07–19.28 µM, respectively. This suggests that the anticancer activities were aided by the addition of small electron-donating groups, such as CH_3_ or OCH_3_, at the *para*-position of the phenyl ring [61]. Abdelgawad et al. (2019) reported the synthesis of 2,4,6-trione **8** in two steps from 3-(1*H*-indol-2-yl)benzenamine **7** (Figure 2) [62]. Compound **8** revealed moderate activity against breast carcinoma (MCF-7), non-small cell lung cancer (A549), human prostate cancer (PC-3), human pancreatic cancer (PaCa-2), and colorectal adenocarcinoma (HT-29) cell lines with IC_50_ values of 4.3–8.8 μM. This result marks hybrid **8** as a target for further development in the field of anticancer agents [62]. Compounds **10a** and **10b** were synthesized from 4-hydroxy-2-mercapto-6- (4-methoxyphenyl)pyrimidine-5-carbonitriles **9** in three steps: reaction with 2-chloromethyl benzimidazole at reflux for 10 h in methanol with the formation of the first benzimidazole–pyrimidine hybrids, synthesis of the chlorinated pyrimidine derivative by reaction with phosphoryl chloride, followed by reaction with amines, morpholine or 2-aminoethanol (Figure 3). As can be seen, the presence of the 2-aminoethanol group on the pyrimidine nucleus in compound **10a** favored a better reaction yield (78%) compared to the presence of the *N*-morpholino group in compound **10b**, resulting in a yield of 70%. The most active compound **10a** exhibited broad-spectrum cytotoxic activity against 25 cancer cell lines, with growth inhibitory activity of 88.84%, 79.89%, and 84.19% against HOP-92 (non-small cell lung cancer), A498 (renal cancer), and T-47D (breast cancer cell line), respectively. Compound **10b** exhibited 61.80% growth inhibitory activity against the MOLT-4 (leukemia) cell line [63]. Reaction of enone **11** with an appropriately substituted *N*-arylguanidinium nitrate **12** and sodium hydroxide, at reflux in propan-2-ol, generated benzimidazole–pyrimidine hybrids **13a**–**13j** with yields of 24–56% (Figure 4). The presence of the morpholino group in compound **13d** considerably improved the reaction yield, to 53%, as did the methoxy group in **13f** (51.5%). The hydroxy group in compound **13g** improved the yield the most, at 56%. The presence of chlorine led to a dramatic decrease in yield to 24% for compound **13i**, while the simultaneous presence of hydroxy and methoxy groups in compound **13j** led to a decrease in yield to 34% compared to **13g** (56%). Hybrids **13a**–**13j** inhibited at least four cancer-related protein kinases, namely Aurora B, PLK1, FAK, and VEGFR2. It should be noted that the most potent protein kinase inhibitor in the series, **13j**, inhibited several cancer cell lines of the NCI panel in submicromolar concentrations, as can be seen in Table 1. The data in Table 1 show that the presence of a small oxygen-containing substituent at position “3” or “4” of the phenyl ring (derivatives **13f**–**13j**) is important for kinase inhibition. Only low action is displayed by compounds with a bigger substituent (**13d**, **13e**) or those without such a substituent (**13a**–**13c**). The decreased inhibitory activity of the derivatives **13d** and **13e** can be explained by the observation that substituents greater than methoxy may interfere with the pillar-like structure generated by Leu59 and Arg136 at the pocket entrance due to the alignment of this pose [64]. Zheng et al. (2013) synthesized a series of thirteen benzimidazole–pyrimidine compounds, having the imidazole ring as the connecting bridge between the two target rings (Figure 5) [65]. All compounds were tested against the cancer cell lines, human acute monocytic leukemia cell line U937, human chronic myeloid leukemia cell line K562, human non-small cell lung cancer A549, and human colon cancer LoVo and HT29, and analyzed for Aurora A/B kinase inhibitory activity in vitro. Compounds **17a**–**17d** showed similar potency to AT-9283 (IC_50_ ranged from 0.40 to 0.6 µM), with IC_50_ values ranging from 0.30 to 0.80 µM, as shown in Figure 5. The good antitumor activity of compounds **17a**–**17e** was correlated with the small volume of the substituent at the “2” position, such as methyl, methylthio, and methyl sulfonyl, of the pyrimidine nucleus [65]. Following a synthesis procedure similar to Shao and Haoran, Abdel-Mohsen and co-workers (2010) reported benzimidazole–pyrimidine carbonitriles **18**–**37** (Figure 2) of potent antitumor activity against 12 cell lines namely, Cervical carcinoma (KB), Ovarian carcinoma (SK OV-3), CNS cancer (SF-268), Non-small lung cancer (NCI H460), Colon adenocarcinoma (RKOP27), Leukemia (HL60, U937, K562), Melanoma (G361, SK-MEL-28), and Neuroblastoma (GOTO, NB-1). The MTT test was employed in accordance with the Mosmann method to determine the anticancer activity. Figure 2 reports the concentration values required to achieve 50% inhibition of the tumor growth of the tested compounds and of the standards (IC_50_ (nM)) [66]. Guo et al. (2013) synthesized benzimidazole–pyrimidine hybrid **38** by the reaction between benzimidazole and 2-(chloromethyl)-6-methyl-4*H*-pyrido [1,2-a]pyrimidin-4-one in DMF and K_2_CO_3_ as a catalyst [67]. Compound **38** activated Pyruvate Kinase M2 (PKM) in Huh7 cells (human liver cancer cell line), with an EC_50_ value of 70 nM, with a novel binding mode to the PKM2 protein (Figure 3) [67]. Certal et al. (2012) reported a new series of benzimidazole–pyrimidone hybrids as potent and selective PI3Kβ inhibitors [68]. Compound **39**, resulting from the reaction between sodium [4-(morpholin-4-yl)-6-oxo-1,6-dihydropyrimidin-2-yl] acetate and *N*-methyl-1,2-phenylenediamine in pyridine with a yield of 55%, showed significant activity and selectivity for PI3Kβ, with an IC_50_ value of 76 nM. The antitumor activity of compound **39** in the human PTEN-deficient PC3 prostate carcinoma tumor model was investigated, xenografted subcutaneously in SCID mice. Hybrid **39** decreased tumor growth by 45% on day 32 at the end of therapy after being given orally to PC3 tumor-bearing mice at a dose of 300 mg/kg twice a day for nine days [68].

Compound **40** was synthesized by the reaction of (*E*)-4-hydroxy-3-((*E*)-(2-methyl- 1*H*-benzo[*d*] imidazol-5-yl)diazenyl)pent-3-en-2-one and guanidine at reflux for 10 h in acetic acid. A moderate cytotoxic activity was determined for hybrid **40** against 60 types of human cancer cell lines, including leukemia, non-small cell lung cancer, melanoma, colon cancer, CNS cancer, ovarian cancer, renal cancer, prostate cancer, and breast cancer [69]. Chen et al. (2013) reported effects 2-(2-(4-(pyrimidin-2-yl)piperazin-1-yl) ethyl)-1*H*-anthra [1,2-*d*]imidazole-6,11-dione **41** on cytotoxicity by MTT assay and repressing hTERT (human telomerase reverse transcriptase) expression activity by SEAP (Secreted Alkaline Phosphatase) assay (Table 2) [70]. Compound **41** affected SEAP expression without significantly affecting the proliferation of treated H1299 cells; therefore, it could selectively repress hTERT expression. The cytotoxic effect of this compound on normal human diploid fibroblasts IMR90 was also determined. Compound **41** showed an IC_50_ value at 100 µM against IMR90, suggesting that it did not affect the overall growth of normal cells [70]. Compounds **45a**–**45e** were achieved in three steps from anthranilic acid **42** with yields of 76–80% (Figure 6). The best result was observed for compound **45a** (IC_50_ = 0.011 μM), which showed a dihydrofolate reductase inhibitory activity (DHFR) comparable or even superior to methotrexate (IC_50_ = 0.02 μM). The presence of electron-withdrawing functional groups in position “4” of the benzene nucleus led to the loss of DHFR activity for molecule **45c** or the decrease of DHFR activity for **45d** (IC_50_ = 0.532 μM). Also, the presence of an electron-donating group -OCH_3_ in compound **45e** led to the loss of DHFR activity. It is noted that compounds **45c**, **45d**, and **45e**, which had chloro, nitro, or methoxy groups at position “4” in the phenyl ring, showed less activity towards DHFR than compound **45a**, which had a simple phenyl ring, and compound **45b** (IC_50_ = 0.034µM), which was fluoro substituted. Using UV–visible and fluorescence spectroscopy, the initial interaction studies of compound **45a** with calf thymus DNA showed that **45a** successfully intercalated with ct-DNA to generate **45a**. DNA complex that is further corroborated by research on ethidium bromide displacement. The binding interactions of compound **45a** with bovine serum albumin (BSA) demonstrated that hydrogen bonds and van der Waals forces played important roles in the strong association of compound 14.BSA [71,72].

To synthesize hybrids **46**–**48**, Ismail et al. (2020) followed a similar process to Zheng [65]. Compound **46** with three methoxy groups (Figure 4) on the phenyl ring proved the strongest antitumor activity, displaying 35 times the activity of cisplatin against colon HT-29 and 25 times its activity against breast MCF-7 cancer cell lines. With GI_50_ values of 6.92 µM against colon cancer, 7.93 µM against melanoma, 7.30 µM against prostate cancer, and 5.57 µM against breast cancer, hybrid **46** also demonstrated broad-spectrum anticancer activity against seven cancer panels, particularly colon, melanoma, prostate, and breast cancers. With only one methoxyl group, compound **47** demonstrated anticancer activity against the ovarian cell line IGROV1, with a growth inhibition (GI) of 59.1%. With a GI of 51.53%, compound **48**, which likewise has three methoxy groups on the phenyl ring connected to pyrimidine, exhibits anticancer action against the leukemia cell line HL60(TB) [73]. Sana et al. (2021) reported the synthesis of benzimidazole–pyrimidine hybrid **52** in two steps, starting from pyrimidin-2-amine **49** and 2-(trifluoromethyl)- benzimidazol-5-amine **50**, through the intermediate hybrid **51**, as seen in Figure 7 [74]. Compound **52** demonstrated the best antitumor activity among a series of synthesized benzimidazole–pyrimidine hybrids. Thus, at concentrations between 2.21 and 7.29 μM, hybrid **52** had the highest cytotoxic activity against the human lung cancer cell line A549. Additionally, **52** demonstrated the most promising anticancer action against the A549 cell line (IC_50_ = 2.21 ± 0.12 μM) by superior microtubule disruption, reduced mitochondrial membrane potential that caused DNA damage, colony-forming ability, and cellular migratory impairment. These characteristics cause tubulin polymerization to be inhibited (IC_50_ = 5.72 ± 0.51 μM) in the G2/M phase, which stops cell proliferation. Structure–activity relationship (SAR) investigations revealed that the hybrids with amine linkage exhibited higher anticancer efficiency against all tested cell types. A sterically hindered trifluoromethyl substituent at the C2-position of the benzimidazole ring, as in compound **52**, enhanced in vitro cytotoxicity against the A549 cell line [74,75].

Rashid et al. (2019) synthesized dione **54** in four steps, starting from 4-oxobutanehydrazide **53** (Figure 8) [76]. Compound **54** displayed remarkable cytotoxic potential against all investigated cell lines, leukemia, colon, melanoma, CNS, prostate, breast, ovarian, renal, non-small cell lung cancer, with GI_50_ values obtained between 0.09 and 16.2 µM falling within the sensitive range. The antiproliferative effects of oxadiazole conjugate **54** were found to be more potent than those of thiadiazole, triazolo-thiadiazines, and triazolo-thiadiazoles, according to the scientists [76]. Docking of hybrid **54** into the enzyme active site yielded a number of molecular interactions showing hydrogen bond, π interactions, and hydrophobic interactions between the drug and enzyme, which are considered to be accountable for the affinity of compound **54**. In the hydrogen bond interaction between the carboxyl group (C=O) of the side chain residue of Arg 364 (1.73 Å) and the nitrogen (–N–) of the imidazole ring of compound **54**, the latter functions as a hydrogen bond acceptor and the former as a donor. Additionally, the carbonyl group (C=O) of hybrid **54** acts as the hydrogen bond acceptor in the second hydrogen bond interaction, and an amino group (N–H) of the side chain residue of Arg 364 (2.39 Å) is a hydrogen bond donor (Figure 5) [77]. Bagul et al. (2023) synthesized a series of benzimidazole-bridged pyrazolo[1,5-a]pyrimidine **57**–**60** by reaction between pyrazolo[1,5-a]pyrimidine-5-carboxylate **55** and substituted benzene-1,2-diamines **56a**–**56d** (Figure 9) [78]. Antiproliferative activity, ranging from 3.2 to 47.4 μM, was observed against panel of cancer cell lines which included MCF-7 (breast cancer), A549 (lung cancer), HeLa (cervical cancer), SiHa (cervical cancer), and significant anticancer activity against cell lines MCF-7, A549, and HeLa (IC_50_ = 3.2–9.3 µM). Also, hybrids **57**–**60** were found to be less cytotoxic to normal lung fibroblast MRC5 cells. The binding pose for hybrid **57** (Figure 6A,B) shows that the molecule binds well in the ATP binding site. The C-2 phenyl ring was buried in the hydrophobic specificity pocket enclosed by Ala719 (1.87 Å), Ile720 (2.99 Å), Lys721 (2.72 Å), Glu738 (4.23 Å), Leu764 (2.25 Å), Ile765 (2.93 Å), and Thr766 (2.16 Å) amino acids. These interactions in the hydrophobic specificity pocket are important for attaining the specificity among the other kinases. The superimposed pose of hybrid **57** with cocrystal ligand erlotinib (Figure 6D) showed that the C-2 phenyl ring of **57** overlapped with the phenylacetylene group of Erlotinib. The C–7 phenyl ring was found oriented towards the hinge region, where 3-methoxy (2.07 Å) and 4-methoxy (3.63 Å) formed hydrogen bonds with the backbone NH of Met769 [78].

This hydrogen bond is vital for EGFR inhibitors and is present in all the drug molecules and ATP as well [79,80,81]. Venugopal et al. reported compound **US10093668B2** that inhibited Mitogen-Activated Protein Kinase (MAPK) and was effective in acute myeloid leukemia (Figure 7) [82]. MAPK-interacting kinases (MNKs) are involved in the phosphorylation of initiation factor 4E (eIF4E), where their protein pathway MNK1/2 eIF4E is overexpressed in cancer cells [83,84]. Dovitinib (TKI-258/CHIR-258), a pan receptor tyrosine kinase (RTK) inhibitor, primarily targets fibroblast growth factor receptor (FGFR), platelet-derived growth factor receptor (PDGFR), vascular endothelial growth factor receptor (VEGFR), fms-like tyrosine kinase 3 (FLT3), and Proto-Oncogene Receptor Tyrosine Kinase (c-KIT) [85]. Dovitinib has demonstrated antitumor activity in pre-clinical models of several cancers and is currently in clinical trials for renal, prostate, adenoid cystic, gastrointestinal, urothelial, thyroid, pancreatic, breast, glioblastoma, and non-small lung cancers [86]. Yadav et al. recently demonstrated the efficacy of Dovitinib in the systemic treatment of mice harboring prostate cancer xenograft tumors [87]. The formation and function of immune cells depend on the protein tyrosine kinase Janus kinase 3 (JAK3), which is involved in cytokine receptor signaling. Immune cell cancers and immunodeficiency can result from aberrant JAK3 activity, especially brought on by mutations. Because cytokine signaling pathways are frequently dysregulated in cancer cells, JAK3 plays a part in cancer [88]. Chen et al. (2006) synthesized hybrid **63** in four steps starting from 4-chloro-2-(methylthio)pyrimidine **61** through the intermediate 2-(methylthio)pyrimidin-4-amine **62** (Figure 10) [89]. Compound **63** showed low nanomolar IC_50_ activity, of 45 nM against JAK3 and of 124 nM against JAK2. The inhibitory activity of hybrid **63** against JAK3 and JAK2 was assessed by the kinase-Glo luminescent assay with Tofacitinib as a reference [89]. Sabat et al. (2006) reported compounds **64** and **65** as inhibitors of lymphocyte-specific kinase (Lck) with IC_50_ values of 193 and 24 nM, respectively [90]. In the same study, the series of compounds **66**, **67**, and **68** showed an excellent potency, with LcK IC_50_ of 3 nM (the key phenolic residue is colored in green, and the 6-pyrimidine substituent essential for anticancer activity is highlighted in yellow in Figure 8) [90]. Mostly present in cells of the myeloid and B-lymphocyte lineages, Hck (hematopoietic cell kinase) is a protein tyrosine kinase that is a member of the Src family. It is essential to the growth and spread of several cancers, as well as signaling pathways that are important in immune cell function, such as cytokine and Fc receptor signaling. Additionally, Hck may be a therapeutic target for Bcr/Abl-chronic myeloid leukemia and HIV infections. To establish the binding mode of these molecules, an X-ray crystallographic structure of **68** with Hck was obtained (Figure 9). Inhibitor **68** orients itself in the Hck enzyme such that the benzimidazole N–H bonds are associated through hydrogen bonds with the amide N–H of Met319, and the phenolic OH hydrogen bonds with the carboxyl of Glu 288. An additional interaction may occur between the O–H of the Thr316 and the aniline N–H on the phenol substituent. The 4-methyl group on the phenol substituent appears to sit in a small hydrophobic groove formed in part by the CH_3_ of Thr316. Thus, the phenolic OH optimally interacts with Glu288 [90]. Hunt et al. (2009) developed a family of benzimidazole–pyrimidine hybrids **69**–**76**, as subnanomolar inhibitors of Lck and also low-nanomolar inhibitors of cellular IL2 release (Figure 10) [91]. As can be seen, compound **69** is a potent inhibitor of both Lck activity and cellular IL2 release. The strongest inhibitor of Lck activity in this series, hybrid **72** (IC_50_ = 0.06 nM), only inhibits cellular IL2 production at a concentration of 111 nM, which represents a nearly 2000-fold shift in potency from the enzyme to the cell. Compounds **69**, **70**, and **71,** with IL2 values of 8, 16, and 42 nM, respectively, showed much less dramatic shifts in potency from the enzyme to the cell, as compared to the parent piperazine **72**, with IL of 111 nM. Hybrid **76**, *N*-ethyl substituted, has a similar behavior to **72**, while hybrids **73**, 74, and **75** have excellent LcK IC_50_ values of 3, 1.5, and 1 nM, respectively [91]. Padhy et al. (2019) reported synthesis of hybrids **77**–**81** (Figure 11) from 2-acetylbenzimidazole in three steps: (1) Claisen–Schmidt condensation of 2-acetylbenzimidazole with substituted aromatic aldehydes in presence of NaOH; (2) nucleophilic substitution of chalcones previously synthesized with benzyl chloride to obtain *N*-benzyl substituted benzimidazole chalcones; and (3) condensation of the previous chalcones with guanidine hydrochloride [92]. The in vitro anticancer activities (cell viability assay) of all compounds were evaluated by SRB assay against the human breast cancer cell line MDA-MB-231 [92]. Compounds **77** (GI_50_ = 84.0 μM) and **78** (GI_50_ = 39.6 μM) exhibited weak activity compared to the standard drug adriamycin (GI_50_ = 0.04 μM). Unexpectedly, the halogeno-substituted compounds **79**, **80**, and **81** showed IC_50_ > 100 µM [92].

## 3. Antimicrobial Benzimidazole–Pyrimidine Hybrids

Events in recent years have demonstrated the need to find new antimicrobial compounds that prevent the unwanted effects that occur with the frequent use of classic antibiotics, as well as antibiotic resistance, and, also, the urgency of more in-depth studies regarding the structure–property relationship in order to generate more effective compounds against a much wider range of pathogens [93]. Research in recent years has shown greater efficacy in antimicrobial compounds with multiple heterocyclic rings, as well as various SAR studies on these hybrids [94]. In the following, we will present studies on benzimidazole–pyrimidine hybrids with antimicrobial properties.

According to Chen et al. (2014), hybrids **86**–**92** were synthesized in three steps from 2-(chloromethyl)-benzimidazole **82** and 2-mercapto-6-methylpyrimidin-4-ol **83** [95]. The intermediate **84** was halogenated with POCl_3_ and PCl_3_ at 110 °C, yielding hybrid **85** (Figure 11) [95]. The reaction of compound **85** with various aromatic amines led to hybrids **86**–**92**, which were tested against four bacterial strains (*Staphylococcus aureus*, *Bacillus subtilis*, *Escherichia coli*, *Stenotrophomonas maltophilia*) and one fungal strain (*Candida albicans*) in vitro, with levofloxacin as a positive control drug for bacterial strains and fluconazole as a positive control drug for fungi. All compounds exhibited inhibitory activity against *S. maltophilia* with MICs ranging from 2 to 32 µg/mL, as shown in Figure 11, and hybrids **86** to **90** exhibited antimicrobial properties against all species of Gram-positive, Gram-negative bacteria, and fungi used in the study. For compounds **86** and **89**, we observe the best antimicrobial activities, which means that the presence of fluorine and methyl substituents in position “4” of the phenyl nucleus was the most beneficial for their bioactivity. The cytotoxic activity of the same compounds showed that only compounds **86** and **89** exhibited enhanced activities against MGC-803 (human gastric cancer cell line), with IC_50_ values of 5.77 µM and 7.39 µM, respectively, compared with 5-Fu (IC_50_ = 8.13 µM) in vitro [95]. Kunduru et al. (2014) synthesized benzimidazole-pyrimidine hybrids **94a**–**94h** by the reaction of substituted 3-phenylpropenones **93a**–**93h** with guanidine, at reflux in acetic acid for 4–5 h (Figure 12) [96]. A maximum reaction yield of 80% is observed for compound **94d** with the methyl group in the “5” position of the benzene ring and the methoxy group in the “4” position of the benzene nucleus. A slightly lower yield (73%) is given by compound **94c** with two methoxy groups grafted onto the benzene nucleus. Antibacterial activity of the synthesized compounds was screened against *B. subtilus*, *S. aureus*, *E. coli*, and *K.* antifungal activity against *F. oxysporum* and *A. niger* (Figure 12). It was noted that compound **94f** has excellent activity against all bacterial strains compared to Steptomicyn and Fluconazole as standards. Hybrid **94b** displayed high activity against *B. subtilus* and good activity against *E. coli* and *K. pneumoniae*, and **94c** shows good activity against all organisms except *E. coli* [96]. Starting from 5-fluorouracil, Fang et al. synthesized a number of benzimidazole–pyrimidine hybrids as a new class of potential antibacterial agents (Figure 12). Compound **97** with 3-fluorobenzyl groups exhibited superior antibacterial activity to chloromycin against *S. aureus*, *B. subtilis*, *E. coli* DH52, and *E. coli* JM109. Additionally, its anti-MRSA activity (MIC = 2 µg/mL) was eight-fold higher than that of chloromycin (MIC = 16 µg/mL) and four-fold higher than that of norfloxacin (MIC = 8 µg/mL). Of all the newly synthesized benzimidazole compounds, compound **97** was the most effective against the Gram-negative bacteria *Bacillus typhi* (MIC = 8 µg/mL). It was found that intermediate **95** displayed good antimicrobial activities against most of the tested bacterial and fungal strains, as seen in Figure 13. With MIC values of 2 µg/mL against *C. albicans* and 8 µg/mL against *S. aureus,* intermediate **95** was found to have strong antibacterial activity against the majority of the tested bacterial and fungal species. Additionally, the di-*n*-pentyl-substituted hybrid **96** has good antibacterial activity against *S. cerevisiae* (MIC = 8 µg/mL) and *B. proteus* (MIC = 8 µg/mL) [97]. AlNeyadi et al. (2017) reported the synthesis of benzimidazole–pyrimidine acrylonitrile hybrids **98**–**100** by reaction between 2-(1*H*-benzo[*d*]imidazol-2-yl)acetonitrile and 2-substituted pyrimidine-5-carbaldehyde in piperidine at 25 °C in 81–89% yields [98]. Hybrid **98** exhibited good antibacterial activity against both Gram-positive and Gram-negative bacteria with a MIC ranging from 9 to 13 µg/mL. Compounds **99** and **100** exhibited moderate antibacterial activity against the tested organisms with MICs of 11.1 to 25 µg/mL (Table 3) [98]. Hessein et. al. (2016) reported the synthesis of hybrid **103** in two steps, from chloroacetamide **101** and quinazoline-2-thiol **102** (Figure 14). Compound **103** had moderate activity against two fungal strains, *Candida albicans* (ATCC 10231) and *Aspergillus fumigatus* [99]. Chikkula and Sundararajan (2017) reported the synthesis of two benzimidazole–pyrimidine hybrids, **105** and **106**, in two steps, starting from benzenamine **104** (Figure 15) [100]. The antibacterial activity of the compounds was evaluated against four Gram-positive bacteria, *Bacillus cereus* ATCC 11778, *Staphylococcus aureus* ATCC 9144, *Micrococcus luteus* ATCC 4698, and *Staphylococcus epidermidis* ATCC 155, and three Gram-negative bacteria, *Klebsiella pneumoniae* ATCC 11298, *Pseudomonas aeruginosa* ATCC 2853, and *Escherichia coli* ATCC 25922. The antifungal activities of the synthesized compounds were evaluated against two fungi, *Aspergillus fumigatus* ATCC 46645 and *Aspergillus niger* ATCC 9029. The antimicrobial activity of compound **105** was, in most cases, twice as good as that of compound **106**, but four to eight times weaker than that of the standard compounds considered, Ciprofloxacin and Ketoconazole, as can be seen in Figure 15 [100]. Saundane and Mathada (2016) reported the synthesis of indole grafted benzimidazole–pyrimidine hybrids **110** and **111** from ethanone **107** and substituted 2-phenyl-1*H*-indole-3-carbaldehydes **108a**–**108c**, via intermediate chalcones **109a**–**109c** (Figure 16) [101]. All compounds had good antimicrobial activity on bacterial strains, *Escherichia coli* MTCC 723, *Staphylococcus aureus* ATCC 29513, *Klebsiella pneumonia* NCTC 13368, and *Pseudomonas aeruginosa* MTCC 1688, using Gentamycin as a reference, and on fungal strains, *Aspergillus oryzae* MTCC 3567T, *Aspergillus niger* MTCC 281, *Aspergillus flavus* MTCC 1973, and *Aspergillus terreus* MTCC 1782, using Fluconazole as a reference, by the serial dilution method, with a MIC ranging from 4 to 16 µg/mL for the best compounds, **110a** and **111a** (Figure 16). It was found that compound **111a**, which had –OH group position “2” of the pyrimidine nucleus and was substituted with chlorine on the indole nucleus, exhibited the highest biological activity [101]. Liu et al. (2018) obtained hybrids **112**–**115** in 58–72% yields, also using an intermediate benzimidazole chalcone (Figure 12) [102]. Compounds **112**–**115** were evaluated for their antimicrobial activities in vitro against four Gram-positive bacteria, *Staphylococcus aureus* ATCC25923, Methicillin-Resistant *Staphylococcus aureus* N315, *Bacillus subtilis*, and *Micrococcus luteus* ATCC4698, six Gram-negative bacteria, *Bacillus proteus* ATCC13315, *Escherichia coli* DH52, *Pseudomonas aeruginosa*, *Bacillus typhi, Escherichia coli* JM109, and *Shigella dysenteriae*, and five fungi, *Candida albicans* ATCC76615, *Candida mycoderma, Candida utilis, Saccharomyces cerevisiae*, and *Aspergillus flavus*, using the standard two-fold serial dilution method in 96-well micro-test plates (Table 4). Based on their antimicrobial activity, 4-methyl-substituted hybrid **112** had the strongest antibacterial activity, followed by 2,4-dichloro-substituted hybrid **113**, 4-fluoro-substituted compound **114**, and 4-chloro-substituted compound **115**, which had the lowest activity. Analyzing hybrid **112**’s antibacterial activity required a flexible ligand receptor docking study. The interaction of compound **112** with the gyrase–DNA receptor is shown in Figure 13. The gyrase residue ASP1083 was in close proximity to the NH group of the benzimidazole ring via hydrogen bonds, with distances of 1.6 Å and 1.9 Å, respectively. The ability of compound **112** to form hydrogen bonds with the residue ASP1083 through the NH_2_ group’s hydrogen atom demonstrated the significance of NH and NH_2_ groups in biological activity. Additionally, the aromatic fragment of compound **112** had electrostatic interactions with the gyrase residues ARG1122, MET1121, ALA1120, TYR1087, and ASP1083. Compound **112** may have strong inhibitory activity against the tested strains due to cooperative binding that may be advantageous for stabilizing the compound–enzyme–DNA complex [102]. Khan et al. (2021) reported the synthesis of hybrids **117**–**124** by the reaction between 2-chloromethylbenzimidazole **82** and 6-substituted tetrahydropyrimidines **116** (Figure 17) [103]. All compounds had extremely strong antimicrobial activity against Gram-negative strains *E. coli*, *P. aeruginosa*, and Gram-positive strains *S. aureus*, *S. pyogenes*, compared to Ampicillin as a standard, and a fungus, *C. albicans*, with Griseofulvin as a reference, as seen in Figure 17. The MIC of compound **117** against *E. coli* was 62.5 µg/mL, which was much more potent than Ampicillin, while hybrids **119**, **121**, and **122** were equipotent with the standard, at 100 µg/mL. *P. aeruginosa* was sensitive to all derivatives at 62.5, 100, and 250 µg/mL, but not to Ampicillin. *Staphylococcus aureus* was sensitive to compounds **117**, **118**, **120**, **121**, and **123**, at 200, 100, 100, 100, and 200 µg/mL, respectively; therefore, they are more potent than ampicillin, with MIC = 250 µg/mL. Compounds **118**–**122** exhibited two-fold greater antifungal activity against *C. albicans* compared to Griseofulvin, with MIC values of 250 µg/mL [103]. Zaghary et al. (2021) reported the synthesis of hybrid **125** by refluxing a mixture of sulfamethazine and (*E*)-1-(1*H*-benzo[*d*]imidazol-2-yl)-3- (dimethylamino)prop-2-en-1-one in acetic acid for two hours (Figure 14) [104]. Compound **125** had antimicrobial activity comparable to standard Ciprofloxacin against the microbial strains *Staphylococcus aureus* ATCC 29213, *B. subtilis* ATCC6633, *B. Cerrus* MTCC 1305, *E. coli* ATCC 2592, and *Pseudomonas aeruginosa* ATCC 27953, considering the average diameter of inhibition zones [104]. Sun et al. (2021) reported the synthesis of 2-(2-alkylthio-6-phenylpyrimidin-4-yl)-1*H*- benzimidazoles **126**–**131** in three steps, from ethanone **107**, namely (1) aldol condensation with different aldehydes in NaOH solution to yield α,β-unsaturated ketones; (2) cyclization of previous ketones with thiourea in sodium isopropyl-isopropanol at reflux to afford substituted pyrimidine-2-thiols; and (3) previous key intermediates reacted with various alkyl halides or benzyl halides under the catalysis of NaOH in acetonitrile, with the formation of hybrids **126**–**131** in yields of 61–87% (Figure 18) [105]. With EC_50_ values ranging from 0.14 to 0.28 μg/mL, compounds **126–131** demonstrated exceptional fungicidal activity against *B. cinerea*, proving that their activities were on par with or higher than carbendazim (EC_50_ = 0.21 μg/mL). Compounds **127**, **129**, **130**, and **131** displayed notable fungicidal activities against *S. sclerotiorum*, with EC_50_ values of 4.65–13.97 μg/mL, which illustrated that their activities were also comparable or higher than that of carbendazim (EC_50_ = 13.32 μg/mL, Figure 18). Structure–activity relationship (SAR) studies revealed that for *S. sclerotiorum*, the antifungal activity towards the R^2^ decreases in the order C_6_H_4_CH_2_ > *n*-C_4_H_9_ > CH_3_ when R^1^ is H, F, or OCH_3_ [105]. The authors chose the *β*-tubulin protein from *B. cinerea* as the biological target for docking, because benzimidazole derivatives used as agricultural fungicides (e.g., carbendazim, thiabendazole, benomyl—Figure 15) are inhibitors of β-tubulin [106]. The amino acid sequence of the *B. cinerea* protein (VERSION: AXO78835.1) was obtained from the NCBI protein database [107], and then the homology modeling was carried out using the *Caenorhabditis elegans* microtubule (PDB: 6e88) as a template [108], where their sequence alignment showed a 78.75% identity. The constructed 3D model of *β*-tubulin protein was used as the receptor, and **130** was used as the ligand to perform the docking study [109], with the result shown in Figure 16. As illustrated in Figure 16, the fluorine atom of the benzene ring may form a hydrogen bond with the OH of Tyr-208 residue at a distance of 3.8 Å, while the NH moiety of the benzimidazole ring may form a hydrogen bond with the carbonyl oxygen atom of the Gln-11 residue at a distance of 2.7 Å. Furthermore, the pyrimidine ring of **130** and the benzene ring of Tyr-222 developed a weak π-π interaction. The *β*-tubulin protein and **130** had a strong affinity because of the two hydrogen bonds and one π-π interaction. This could explain the superior fungicidal activity of compound **130**. Vlasov et al. (2021) reported the synthesis of benzimidazole–pyrimidine acetamide **132** by reaction between 6-(1*H*-benzo[*d*]imidazol-2-yl)-3,5-dimethyl-2-thioxo-2,3-dihydro thieno[2,3-d]pyrimidin-4(1*H*)-one and 2-chloro-N-(4-isopropylphenyl)acetamide in dimethyl formamide for 3.5 h [110]. Hybrid **132** was more active than the Streptomycin standard against Gram-positive bacteria, *S. aureus* ATCC 25923, *B. subtilis* ATCC 6633, and demonstrated similar activity to the reference against Gram-negative bacteria, *E. coli* ATCC 25922, *P. vulgaris* ATCC 4636, *P. aeruginosa* ATCC 27853, and also inhibited the growth of the *C. albicans* strain, as evidenced by the average diameter (mm) of the growth inhibition zone [110].

## 4. Antiviral Benzimidazole–Pyrimidine Hybrids

A number of significant viral infections have surfaced in recent years, and the lack of effectiveness of antiviral chemotherapeutic medicines has resulted in severe human illnesses and death. Therefore, finding novel antiviral candidates is the aim of researchers. Because of their wide range of therapeutic benefits, heterocyclic compounds—and more recently, heterocyclic hybrids—have become more significant in the field of medicinal chemistry [111]. The following will outline the antiviral properties of benzimidazole–pyrimidine hybrids. In order to compare the efficacy of pyrimidine–benzimidazole hybrids **117**–**124** (Figure 17) in inhibiting the major protease of SARS-CoV-2 and the receptor binding domain (RBD) of spike glycoprotein with approved drugs and native ligands such as Ivermectin or Favipiravir, Khan et al. (2021) used molecular docking [103]. The binding affinity, the number of hydrogen bonds, and active amino acids of several derivatives, **119**–**124**, were similar to those of approved drugs, as can be seen in Table 5 [103]. The formation of hydrogen bonds with target molecules results in inhibition, but binding affinity can be increased by van der Waals forces π–π, and hydrophobic interactions [103].

El Diwani et al. (2014) synthesized a series of new benzimidazole–pyrimidines **134**–**136** by the reaction between tetrahydropyrimidine-5-carbonitriles **133a**–**133c** and 2-chloromethylbenzimidazole **82** in tetrahydrofuran for 24 h, and the hybrids **137**–**139** by the hydrolysis of the nitriles at reflux in 80% H_2_SO_4_ followed by neutralization with ammonia solution (Figure 19) [112]. All compounds were evaluated for their hepatitis C virus (HCV) RNA replication-inhibitory activity. Compounds **134**–**136** were found to be more potent than VX-950, with IC_50_/IC_90_ of 0.123/0.321 for **134**, 0.145/0.345 for **135**, 0.129/0.432 for **136**, and 0.20/0.45 μM for VX-950 (Telaprevir), respectively (Figure 19). Compound **137**, with IC_50_/IC_90_ of 0.116/0.452 μM, displayed activity very similar to that of the standard. Therefore, compounds **134**–**136** were potent (HCV) RNA replication inhibitors and are good drug candidates for further investigations [112]. By refluxing a mixture of benzimidazole, 4-(4,6-dimethylpyrimidin-2-ylamino)benzene sulfonamide, and formaldehyde for 3 h, Selvam et al. (2010) synthesized benzimidazole–pyrimidine sulfonamide **140** (Figure 17) [113]. The anti-HIV activities were also screened for in vitro antiviral activity against replication of HIV-1 and HIV-2 in MT-4 cells using Zidovudine as a standard, and cytostatic activity was also studied by MT-4/MTT assay. Compound **140** inhibited the replication of HIV-1 and HIV-2 with an EC_50_ of 35.40 μg/mL and a CC_50_ > 125 μg/mL in MT-4 cells, compared to an EC_50_ of 0.0012 and 0.00016 μg/mL, and, respectively, a CC_50_ of 62.40 μg/mL for the standard Zidovudine [113].

## 5. Anti-Inflammatory Benzimidazole–Pyrimidine Hybrids

One of the most often prescribed medications in the world is non-steroidal anti-inflammatory drugs, or NSAIDs. The majority of anti-inflammatory medications currently used in clinical settings are becoming outdated because of their possible negative effects. They are turning out to be very dangerous to utilize for extended periods of time. As a result, new anti-inflammatory drugs have been created recently, and many of them are currently in advanced clinical testing. Benzimidazole–pyrimidine compounds with anti-inflammatory activities reported in the literature are presented below [114]. Chikkula and Sundararajan (2017) studied the anti-inflammatory activity of compounds **105** and **106** (Figure 15) [100]. The carrageenan-induced paw edema test was performed using Wistar rats, and the obtained results are shown in Figure 18. The anti-inflammatory activity of thio-compound **106** was slightly better than that of oxo-compound **105**, and both were half as effective as the standard Diclofenac [100]. Prajapat and Telasara (2016) reported the synthesis of a series of benzimidazole–pyrimidine **142**–**150** compounds containing alkoxy phthalimide [115]. Thus, by the reaction of isoindoline-1,3-dione **141** with benzimidazole–pyrimidine hybrids **142**, **143a**–**143c**, and **144a**–**144b**, the target hybrids **145**–**150** resulted (Figure 20) [115]. The best yield of 62% correlates with the presence of the carboethoxy group in the “5” position of the pyrimidine ring in compound **143b**, while for compound **143c** with the carbomethoxy group in the same position, the yield decreased to 56%. The presence of the acetyl group in the same position of the pyrimidine ring in **143a** led to a yield of 60%. The hybrids **145**–**150** showed significant anti-inflammatory activity against carrageenan-induced rat paw edema. On screening, compounds **145**, **146**, and **149** exhibited good anti-inflammatory activity, whereas compounds **147**, **148**, and **150** exhibited moderate anti-inflammatory activity when compared to the reference diclofenac (Figure 20) [115]. By blocking the activity of lymphocyte-specific kinase (Lck), hybrid **68** demonstrated a strong anti-inflammatory effect, according to Sabat et al. (Figure 8) [90,116]. Compound **68** exhibited strong anti-inflammatory properties and was effective at 0.054 mM for IL-2 cytokine inhibition and 3 nM for Lck kinase inhibition [90].

Chen et al. (2006) found hybrid **63** (Figure 10) as a potent inhibitor of Janus kinase 3 (JAK3) with an IC_50_ activity of 45 nM [89]. SAR studies revealed that the nitrile group at the benzimidazole’s “6” position had a role in the high level of JAK3 inhibition [89]. Hybrids **69**–**76** were identified by Hunt et al. (2009) as inhibitors of Lck and inhibitors of cellular IL2 release (Figure 10) [91]. Compound **69** exhibits strong inhibition of cellular Lck activity and IL2 release, with IC_50_ values of 0.12 and 8 nM, respectively. Hybrid **72**, the most potent Lck activity inhibitor in this series (IC_50_ = 0.06 nM), barely stops cells from producing IL2 at 111 nM. Also, hybrids **73**, **74**, and **75** have strong LcK IC_50_ values of 3, 1.5, and 1 nM, respectively [91]. Abdelgawad et al. (2019) reported the anti-inflammatory activity of compound **8** by determining the inhibitory activity of cyclooxygenase (COX-1, COX-2) and phospholipase A2-V (sPLA2-V) [62]. The results showed that hybrid **8** has a strong inhibitory activity against COX-1 (IC_50_ = 2.76 μM), a moderate activity against COX-2 (IC_50_ = 7.47 μM), and a moderate inhibitory activity against secretory phospholipase A2-V (sPLA2-V), with an IC_50_ of 7.51 μM [62].

## 6. Analgesic Benzimidazole–Pyrimidine Hybrids

The International Association for the Study of Pain (IASP) defines pain as an unpleasant emotional and sensory experience connected to actual or potential tissue damage. It is unethical and morally wrong to ignore pain [117,118]. Chikkula and Sundararajan (2017) reported analgesic activity of compounds **105** and **106** [100]. Compounds generally exhibited moderate analgesic activity at 30 min of reaction time; activity rose in the first hour, peaked in the second, and then declined in the third (Figure 19). These compounds exhibited a mild analgesic response compared to the standard drug Diclofenac [100].

## 7. Antiulcer Benzimidazole–Pyrimidine Hybrids

A chronic condition that affects up to 10% of people worldwide is peptic ulcer disease. Peptic ulcers are caused by a decline in mucosal defenses and the presence of gastric juice pH. The two main causes of the disruption of mucosal resistance to injury are *Helicobacter pylori* (*H. pylori*) infection and non-steroidal anti-inflammatory drugs (NSAIDs). Both the benzimidazole nucleus and the pyrimidine nucleus are known from the literature to have antiulcer properties [14,119,120]. The synthesis and antiulcer properties of the benzimidazole–pyrimidine hybrids will be discussed below. Mathew et al. (2013) reported the synthesis of 2-substituted benzimidazole-pyrimidine-2,4,6-triones **157**–**162** by refluxing a mixture of enone **150**–**155** and pyrimidine-trione **156** in acetic acid (Figure 21) [121]. Compounds **158**, **160**, and **159** showed the best percentage protection of 69.58, 69.56, and 67.17%, respectively, at a dose of 50 mg/kg b.w. when compared to the standard omeprazole (77.37%, 2 mg/kg body weight). The total acidity and pH values from Figure 21 revealed that hybrids **157**–**162** had a good protective effect on the gastric mucosa in ulcers. The mice treated with each of the hybrids **157**–**162** showed no damage to the stomach mucosa, according to electron microscope scanning of stomach specimens. Hence, each derivative demonstrated good mucomembranous protection [121]. Farhan and Farooqui (2021) reported synthesis of benzimidazole–sulfinyl–pyrimidines **166**–**174** by the reaction of 5-substituted benzimidazole-2-thiols **163a**–**163c** with 6-substituted 4-chloro-2-methyl pyrimidines **164a**–**164i** in basic medium of sodium hydroxide, and oxidation of intermediates **166a**–**166i** with *meta*-chloroperoxybenzoic acid (MCPBA) (Figure 22) [122]. All the compounds showed 30 to 70% inhibition of ulcers. It was found that the compounds **166** (45.20%), **169** (41.20%), and **173** (37.10%) exhibited lowest antiulcer activity, while the compounds **167** (51.99%) and **170** (52.68%) showed slightly significant antiulcer activity compared to the standard Pantoprazole (76.16}, and compounds **168** (74.03%), **171** (72.87%), and **174** (75.15%) showed highly significant antiulcer activity compared to the standard drug. The ulcerogenic activity increases in the following order: **173** < **169** < **166** < **167** < **170** < **172** < **171** < **168** < **174** < Pantoprazole. This means that the presence of the *n*-propyl group at the “5” position of the benzimidazole ring is essential for the high ulcerogenic activity, as in the compounds with the best activities, **174**, **168**, and **171** [122]. Patil et al. (2010) used a similar synthetic route to obtain compounds **175**–**181** (Figure 20) [123]. Compounds **178** and **180** showed the most potent activity as compared to Omeprazole at the dose level of 10 and 20 mg/kg, while compounds **175**, **176**, **177**, **179**, and **181** showed moderate activity at the same doses [123].

## 8. Antidepressant Benzimidazole–Pyrimidine Hybrids

According to the World Health Organization, depression is the primary cause of disability globally, affecting an estimated 300 million people. Furthermore, the presence of depression raises the risk of various illnesses like epilepsy, Alzheimer’s disease, stroke, cardiovascular disease, etc., considerably [124]. Finding new antidepressant compounds that better satisfy the increasingly varied needs that emerge globally is therefore crucial. A complex enzyme system in the central nervous system, monoamine oxidase (MAO) specifically catalyzes the deamination or inactivation of biogenic amines. MAO inhibitors increase the intracellular concentration of endogenous amines by inhibiting their deamination, which seems to be the cause of their antidepressant action. Mathew et al. (2016) reported the antidepressant activity of compounds **157**–**162** (Figure 21) as MAO inhibitors [125]. All compounds showed good antidepressant activity when compared to the standard Clomipiramine at a dose level of 20 mg/kg (Table 6). The compound **160** had the best MAO-A inhibitory activity, significantly reducing the duration of immobility times at a 50 mg/kg dose level when compared to the standard Clomipiramine. The compounds **160**, **159**, and **162** significantly reduced the duration of immobility times to 62.58%, 57.23%, and 55.35% at a 50 mg/kg dose level when compared to the standard drug. The results revealed that the electron-donating groups, such as dimethyl amino, methoxy, and hydroxyl groups in the phenyl nucleus of the compounds, significantly enhanced the antidepressant activity when compared to (1*H*-benzimidazol-2-yl)-3-phenylprop-2-]pyrimidine-2,4,6 (1*H*,3*H*,5*H*)-triones having no substituents or electron-withdrawing groups on the phenyl ring system. According to molecular docking studies, the phenyl ring’s high lipophilic group may have an additional hydrophobic binding area that significantly increases the pharmacological action of the CNS antidepressant [125].

Corticotropin-releasing factor 1 (CRF1) receptor antagonists are promising targets for stress-related disorders. Kojima et al. (2018) reported compound **182** as a CRF1 receptor antagonist with IC_50_(binding) of 110 nM and IC_50_(functional) of 210 nM for the racemate [126]. Also, they determined that enantiomer **182**-*R* successfully binds CRF1 receptors in the brain and exhibits the potential to be further examined for clinical studies (Figure 21) [126].

## 9. Anti-Alzheimer’s Benzimidazole–Pyrimidine Hybrids

JNKs, or c-Jun N-terminal kinases, were first discovered to be kinases that phosphorylate and bind to c-Jun on Ser-63 and Ser-73 within its transcriptional activation domain. The three genes JNK1 (four isoforms), JNK2 (four isoforms), and JNK3 (two isoforms) produce the ten isoforms of the c-Jun N-terminal kinases. The JNK3s are regarded as degenerative signal transducers in pathological conditions. During the last two decades, a number of reports have supported JNK as a good therapeutic target for neurodegenerative diseases such as Alzheimer’s and Parkinsonian diseases, in addition to ischemic injury. The direct toxicity of β-amyloid contributes to the neuronal dysfunction and loss observed in Alzheimer’s disease. β-Amyloid-induced cell death is attenuated in cortical neurons from JNJ3-null mice, and JNK3 mediates this cell death through the activation of c-Jun and the enhanced expression of Fas ligand, a protein that plays a role in programmed cell death (apoptosis) and has been implicated in the pathogenesis of Alzheimer’s disease (AD). Kim et al. (2013) reported benzimidazole–pyrimidine hybrids **183**–**185** as inhibitors of c-Jun N-terminal kinases, JNK3 [127]. The synthesis of hybrids **183**–**185** has as an essential step, the reaction between 5-substituted 2-(naphthalen-2-yl)-1*H*-benzo [*d*]imidazoles and 4-chloro-2-(methylthio)pyrimidines on a palladium catalyst, in toluene with the formation of benzimidazole–pyrimidine hybrids [127]. Their activities were evaluated through measurement of K_d_ using SPR (Surface Plasmon Resonance), JNK3 kinase assay, and cell viability of human neuroblastoma cells. Compounds **183**, **184**, and **185** showed strong affinities to JNK3 of 0.743, 1.17, and 0.0461 µM, respectively (Figure 22). The best compound, **185**, was confirmed as a potent and selective JNK3 inhibitor in cells, dramatically reducing phosphorylation of c-Jun. Thus, a dose-dependent decrease in tumor necrosis factor TNFα-mRNA levels by hybrid **185** with an IC_50_ of 1.09 µM was observed in the study [127].

## 10. Antioxidant Benzimidazole–Pyrimidine Hybrids

Antioxidants are substances that can prevent or reduce harm to cells. These molecules are produced by the body in response to environmental stressors and other factors. These compounds, also known as “free-radical scavengers,” are necessary for cells to survive in their internal processes. Antioxidant supplements may help reduce age-related glaucoma-related vision loss in older adults [128].

Abdelgawad et al. (2019) evaluated the free radical scavenging effect of hybrid **8** using a colorimetric test, the DPPH (2,2-diphenyl-1-picrylhydrazyl) radical scavenging test with trolox as a standard [62]. The results of inhibitory effects at different concentrations of 10, 50, and 100 µM, respectively, showed that compound **8** exhibited a good scavenger effect against the DPPH radical [62].

## 11. Solubility of Benzimidazole–Pyrimidine Hybrids

The solubility of the compounds is essential for further pharmacological studies, pharmacokinetic properties, and bioavailability of the compounds. In general, the aqueous solubility of these hybrids is particularly poor. Certal (2012) studied benzimidazole– pyrimidine hybrids **186**–**190** with surprisingly low aqueous solubility, less than 50 µM, at pH 7.4 (Figure 23) [68]. This solubility was correlated with strong crystal packing with intermolecular hydrogen bonds and hydrophobic interactions [68]. Guo et al. (2013) report anticancer hybrids **38**, **191**–**194** with good aqueous solubility of 178–586 µM at pH 7.4 (Figure 24) [67]. In general, small substituents such as methyl, chloro, or methoxy groups improved solubility and also their potency as PKM2 activators [67]. A more detailed study on the solubility of benzimidazole–pyrimidine hybrids was carried out by Sabat et al. [90]. The researchers focused on improving the poor aqueous solubility of compound **64** (solubility of 4 µg/mL) by functionalizing it. The synthesized hybrid **67** had an even poorer solubility of 3 µg/mL (Figure 25). Methylation of its phenolic group improved the solubility nine-fold in hybrid **195** (27 µg/mL). The best solubility in this series was found for compound **68** (127 µg/mL) substituted in the 6-position with (*N,N-*diethylamino) ethylamino, followed by compound **196**, (4-methylpiperazinyl)ethoxy substituted with a solubility of 70 µg/mL. The presence of the secondary amino group in position “6” of the pyrimidine ring and a hydrocarbon chain attached to it was found to be essential for the improved solubility of the compounds **68**, **196**–**199** (Figure 26). In conclusion, from this series of compounds, hybrids **68**, **196**–**199**, had good solubilities of 40–127 µg/mL, and could be further studied for intrinsic clearance and bioavailability [90]. In most cases, the increase in the solubility of the hybrids is accompanied by a decrease in therapeutic activity.

## 12. Current Challenges and Future Prospects

Current challenges in the synthesis of benzimidazole–pyrimidine hybrids include the following:

1. Optimization of reaction conditions for high yields and selectivities. From this review, many syntheses with modest yields of 10–20% or even lower are observed. Therefore, finding new strategies to generate these hybrids is of utmost importance. Studies report various reaction conditions for optimizing yields, such as temperature, solvent, catalyst, and synthesis strategies. A very important role in a synthesis is played by the substituents grafted onto molecules, in the benzimidazole, pyrimidine, or other nuclei present, both through the electronic effects they generate and through the variation in solubility depending on the presence of one or more substituents.

From the presentation of the syntheses of benzimidazole–pyrimidine hybrids, it is observed that the presence of certain substituents, such as hydroxy, thiol, amino, methyl, methoxy, ethoxy, fluorine, trifluoromethyl, led to higher yields compared to molecules that did not contain these substituents. The yield of the synthesis reactions of benzimidazole hybrids is significantly improved by the presence of fluorine, chlorine, amino, methyl, and trifluoromethyl substituents on the benzene ring, while the presence of the same halogen on a phenyl ring linked to one of the rings leads to a substantial decrease in the yield. Methyl, amino, aminophenyl, thiomethyl, and sulfinylmethyl groups directly linked to the pyrimidine ring lead to an increase in the reaction yield. The presence of a cyano group on the benzimidazole ring leads to a strong decrease in the yield. The simultaneous presence of cyano and amino groups on the pyrimidine ring leads to a dramatic decrease in the yield. Hydroxy or morpholino groups, grafted on a phenyl ring, improve the reaction yield, while the simultaneous presence of hydroxy and methoxy groups leads to a dramatic decrease in the yield.

2. Addressing solubility issues for most hybrids, since, as shown above, in general, the problem of aqueous solubility of hybrids is essential, especially in therapeutic formulations. To obtain more water-soluble hybrids, it is necessary to functionalize the rings present in the molecule, both benzimidazole and pyrimidine, or other rings present, and among the substituents that help in this regard, those with amino, hydroxy, ether, or alkylyl, methyl, ethyl, or propyl groups stand out.

3. Understanding structure–activity relationships (SARs) to increase therapeutic potential. Many of the studies mentioned try to achieve a correlation between the architecture of a molecule, the presence of certain substituents, and therapeutic activity. The conclusions are not always uniform. For example, if initially an improved biological activity is noted by the presence of a halogen, fluorine, chlorine, bromine, or iodine, in many of the studies, there is no improvement in the therapeutic activity of benzimidazole–pyrimidine hybrids by their presence. Most likely, in this case, the presence of a halogen leads to a decrease in its solubility, not to a therapeutic improvement. Methylene, sulfidomethylene, amino, diazo, or carbonyl bridges between the two benzimidazole and pyrimidine rings typically increase the biological activity of anticancer drugs, according to SAR studies. It has been demonstrated that the presence of saturated heterocyclic substituents, such as morpholine or piperazine, as well as methoxy and methyl groups on a heterocycle or phenyl ring, is very advantageous for a variety of anticancer compounds. The S=O group between the two cores of pyrimidine and benzimidazole, the grafted alkyl groups—particularly the n-propyl to the pyrimidinone nitrogen—and the substituents -OCH_3_ and -OCF_3_ from the “5” position of the benzimidazolic ring are noted as being crucial for enhanced therapeutic activity. Similar to anticancer compounds, the antibacterial action of compounds is enhanced when a linker, such as methylene, thio, amino, or another heterocycle, is present between the two benzimidazole and pyrimidine rings. SAR studies showed that the high degree of JAK3 inhibition was caused by the presence of the nitrile group at the “6” position of the benzimidazole.

4. Computational methods such as DFT are also used to predict and optimize molecular properties and biological activity. The design of hybrid molecules with therapeutic properties through density functional theory is a widely used method, with good results. The receptor–ligand interaction establishes the active centers of the target molecule to be synthesized, and thus, the functional groups and their positions. Of particular importance in this regard are the hydrogen bonds that are established between the hydroxy or amino groups grafted on the hybrids and the amino acid residues in the receptor protein. More hydrogen bonds mean a stronger interaction and a superior therapeutic activity.

5. The role of chalcogens both within and outside the ring could be beneficial in improving the therapeutic activity through the generated chalcogen–receptor interaction. Chalcogens can be easily bound or assimilated by certain receptor structures due to their similarity to important biochemical structures, such as enzymes.

6. Future prospects focus on improving the efficacy, bioavailability, and safety of the compounds for the development of new pharmaceutical products with superior qualities to those currently known. In this sense, all the aspects discussed above are included here, especially the finding of hybrids with better aqueous solubilities and pharmaceutical formulations that would enhance their therapeutic action. The aim is to design, project, synthesize, and test the compounds and improve their characteristics by structural modification, respectively, grafting of substituents marked to be beneficial for a potential therapeutic activity.

## 13. Conclusions

In this review, the importance of benzimidazole–pyrimidine hybrids was highlighted by discussing various schemes for obtaining them, the tested medicinal properties, as well as the SAR studies performed. It is observed that most of the presented hybrids are potential anticancer agents, with remarkable properties, constituting future drugs to treat different types of cancer. SAR studies for anticancer compounds show that the presence of methylene, sulfidomethylene, amino, diazo, or carbonyl bridges between the two benzimidazole and pyrimidine rings generally improves their biological activity. Also, from the docking studies, it was seen that the presence of an additional heterocycle, such as 1,3,4-oxadiazole, considerably improved the anticancer activity of the hybrids. The presence of methoxy and methyl groups on one heterocycle or phenyl ring on various anticancer molecules has proven particularly beneficial, as has the presence of saturated heterocyclic substituents, such as morpholine or piperazine. The IC_50_ values for many of the reported anticancer hybrids are in the nanomolar range, which signifies particularly potent anticancer agents. The antimicrobial activity of the presented hybrids was particularly effective against a very wide range of pathogens, Gram-positive, Gram-negative bacteria, and fungi, with very good values of the minimum inhibitory concentrations, in the order of µM or µg/mL. SAR and docking studies have shown for many cases a more efficient interaction of antimicrobial molecules when they have electron-withdrawing substituents, such as F, Cl, Br, or OH, grafted onto the aromatic nuclei. These results are consistent with SAR studies reported in the literature for other heterocyclic compounds [129,130,131]. As in the case of anticancer molecules, the presence of a linker, methylene, thio, amino, or another heterocycle between the two benzimidazole and pyrimidine rings improves the antimicrobial activity of the compounds. The hybrids also had a nice range of antiviral activities against SARS-CoV-2, hepatitis C virus (HCV), and HIV-1. The benzimidazole–pyrimidine hybrids had very good anti-inflammatory activities, with IC_50_ values in the nanomolar and subnanomolar range. A series of compounds were JAK3 inhibitors and inhibitors of cellular IL2 release. SAR studies revealed that the nitrile group at the benzimidazole’s “6” position had a role in the high level of JAK3 inhibition. Compounds with strong inhibitory activity against COX-1 were also reported. Antiulcer activity of the hybrids discussed was close to the benzimidazole-containing standards, omeprazole and pantoprazole. It is highlighted the importance of the S=O group between the two cores benzimidazole and pyrimidine, as well as of the grafted alkyl groups, especially of the *n-*propyl to the pyrimidinone nitrogen, and of the substituents -OCH_3_ and -OCF_2_ from the “5” position of the benzimidazolic ring. The antidepressant activity of these compounds is remarkable. A number of compounds were MAO-A inhibitors and CRF1 receptor antagonists. According to molecular docking studies, the phenyl ring’s high lipophilic group may have an additional hydrophobic binding area that significantly increases the pharmacological action of the CNS antidepressant. The benzimidazole–pyrimidine hybrids have been shown to be selective JNK3 inhibitors in cells, and thus are potential anti-Alzheimer’s agents. The analgesic and antioxidant properties of these hybrids are also reported. These encouraging results motivate more research in this area. Many of the findings are only the start of future research directions that could transform medicinal chemistry. For upcoming generations of researchers, chemists, pharmacists, and biochemists, we hope this review will be a helpful resource.

## Data Availability

No new data were generated or analyzed in support of this research.

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
