# Peer review of "Benzimidazole–Pyrimidine Hybrids: Synthesis and Medicinal Properties"

_pharmaceuticals, 2025, doi:10.3390/ph18081225_

Round 1

Reviewer 1 Report

Comments and Suggestions for Authors

The present review, titledBenzimidazole-Pyrimidine Hybrids: Synthesis and Medicinal Properties presents a comprehensive overview of recent advances on  synthesis strategies of benzimidazole-pyrimidine hybrid and their promising biological properties. The paper is generally well structured and covers a wide range of synthetic strategies However, the manuscript requires minor revision before it can be considered for publication.

1/ The manuscript needs significant editing as there is a large number of typographical and  grammatical errors.

2/ In abstract and the sentence in line 74 that list various websites (ACS Publications, PubMed, MDPI, etc.) should be revised. Instead of naming specific databases or publishers, a general phrase such as "relevant scientific literature from peer-reviewed sources" would be more appropriate and professional in a review article.

3/In its current form, the manuscript includes a large number of tables—some of which are redundant or not absolutely necessary. To improve fluidity and avoid visual overload, I recommend removing certain tables and instead integrating key ICâ‚…â‚€ values directly into the chemical schemes, in parentheses next to the corresponding structures. This would make the figures more informative while simplifying the overall layout.

Author Response

The present review, titled “Benzimidazole-Pyrimidine Hybrids: Synthesis and Medicinal Properties presents a comprehensive overview of recent advances on  synthesis strategies of benzimidazole-pyrimidine hybrid and their promising biological properties. The paper is generally well structured and covers a wide range of synthetic strategies However, the manuscript requires minor revision before it can be considered for publication.

Comments 1/ The manuscript needs significant editing as there is a large number of typographical and  grammatical errors.

Response 1: Thank you for your observation. We have checked the manuscript and made the necessary corrections.

Comments 2/ In abstract and the sentence in line 74 that list various websites (ACS Publications, PubMed, MDPI, etc.) should be revised. Instead of naming specific databases or publishers, a general phrase such as "relevant scientific literature from peer-reviewed sources" would be more appropriate and professional in a review article.

Response 2: Thank you for poiting this out. I wrote at your suggestion: "A literature search was conducted across relevant scientific literature from peer-reviewed sources,"

Comments 3/In its current form, the manuscript includes a large number of tables—some of which are redundant or not absolutely necessary. To improve fluidity and avoid visual overload, I recommend removing certain tables and instead integrating key ICâ‚…â‚€ values directly into the chemical schemes, in parentheses next to the corresponding structures. This would make the figures more informative while simplifying the overall layout.

Response 3: Thank you for this guidance. Of the 21 tables in the original manuscript, only 6 tables remain. The rest of the data has been given either in schemes or on figures.

Reviewer 2 Report

Comments and Suggestions for Authors

 Both benzimidazoles and pyrimidines are essential heterocycles in medicinal chemistry, with various therapeutic properties. The authors summarizes benzimidazole-pyrimidine hybrids with diverse medicinal properties, and presents synthetic methods. The manuscript will be a good guidance for the researchers in this field.

The following issues should be addressed.

  1. The abstract is too lengthy, it should be significantly reduced and rewritten.
  2. In the Introduction, did anyone revewe the synthesis and medicinal Properties of benzimidazoles or pyrimidines? This should be mentioned in the paragraph across lines 69-70..
  3. Lines 77-79 should be deleted.
  4. Through the manuscript, compound numbers should be bold.
  5. The IUPAC names of compounds emerge through the manuscript, which reduces the readibility. Since compound numbers are given, the authors should mention the key structural skeletal that represents the strongest novelty of the cited work and the compound numbers, instead of their IUPAC names. For example, in line 139, "3-(N,N-Dimethylamino)-1-(1-methyl-1H-benzimidazol-2-yl)prop-2-en-1-one 11" can be abbreviated as  "the substituted enone 11".
  6.  in line 173, "2-((1H-benzo[d]imidazol-2-yl)methylthio)-4-(substituted)-6-phenylpyrimidine-carbonitrile 18–37". Abdel-Mohsen and co-workers incorporated a thioether substrcture and a cyano group to obtain a new class of benzimidazole-pyrimidine hybrids 18–37. The thioether and the cyano group are their structural novelties. Similar problems are omnipresent in the manuscript.
  7. The manuscript can be published after the authors adress the above concerns.

Author Response

 Both benzimidazoles and pyrimidines are essential heterocycles in medicinal chemistry, with various therapeutic properties. The authors summarizes benzimidazole-pyrimidine hybrids with diverse medicinal properties, and presents synthetic methods. The manuscript will be a good guidance for the researchers in this field.

The following issues should be addressed.

  1. The abstract is too lengthy, it should be significantly reduced and rewritten.

Response 1: Thank you for your observation. According to the requirements of the Pharmaceuticals journal, the Abstract should contain around 250 words. Our Abstract contained 272 words, and now, after we rewrote it, it contains 244 words.

  1. In the Introduction, did anyone revewe the synthesis and medicinal Properties of benzimidazoles or pyrimidines? This should be mentioned in the paragraph across lines 69-70..

Response 2: Yes, in the Introduction, in lines 59-64, it is written about the medicinal properties of benzimidazoles and pyrimidines:

"In this review, starting from the fact that both the pyrimidine [18-20] and the benzimidazole [21-23] nuclei are found as key nuclei in drugs with various therapeutic applications, such as, anticancer [24,25], antibacterial [26,27], antifungal [28,29], antiviral [30,31], antidiabetic [32-34], antiulcer [35,36], antioxidant [37,38], anti-Alzheimer [39,40], antidepressant [41,42], anti-Parkinson [43,44], anticonvulsant [45,46] anti-inflammatory [47,48],"

For the synthesis methods for benzimidazole compounds, we added the paragraph:

"Recent articles indicate various routes for the synthesis of benzimidazoles, such as the coupling of 1,2-diaminobenzenes with carboxylic acids (Phillips-Ladenburg reaction), the coupling of 1,2-diaminobenzenes with aldehydes and ketones (Weidenhagen reaction) or the rearrangement of quinoxalinones [14,16]."

And for the synthesis methods of pyridines, the paragraph below:

Various methods for the synthesis of pyrimidines are studied in the literature, such as, two-component cycloadditions, like, the [5+1] annulation of enamidines, [4+2] cycloadditions, [3+3] cycloadditions; three-component cycloadditions or the Biginelli reaction [18,19].

  1. Lines 77-79 should be deleted.

Response 3: Thank you for your observation. I deleted lines 77-79

  1. Through the manuscript, compound numbers should be bold.

Response 4: We have checked the entire manuscript and have noted all compounds in bold.

  1. The IUPAC names of compounds emerge through the manuscript, which reduces the readibility. Since compound numbers are given, the authors should mention the key structural skeletal that represents the strongest novelty of the cited work and the compound numbers, instead of their IUPAC names. For example, in line 139, "3-(N,N-Dimethylamino)-1-(1-methyl-1H-benzimidazol-2-yl)prop-2-en-1-one 11" can be abbreviated as  "the substituted enone 11".

Response 5: Thank you for your observation. It is indeed easier to follow the article if it is named as a function and the number of the compound. I have checked the whole article and renamed the necessary places.

Page 3: Abdelgawad et al. (2019) reported synthesis of 2,4,6-trione 8 in two steps from 3-(1H-indol-2-yl)benzenamine 7 (Scheme 2).

Page 4: Reaction of enone 11 with an appropriately substituted N-arylguanidinium nitrate 12 and sodium hydroxide, at reflux in propan-2-ol generated benzimidazole-pyrimidine hybrids 13a13j with yields of 24-56% (Scheme 4). Hybrids 13a13j....

Page 5: Abdel-Mohsen and co-workers (2010) reported benzimidazole-pyrimidine carbonitriles 1837 (Fig. 2) of potent antitumor activity against 12 cell lines namely...

Page 7: Sana et al. (2021) reported synthesis of benzimidazole-pyrimidine hybrid 52 in two steps, starting from pyrimidin-2-amine 49 and 2-(trifluoromethyl)- benzimidazol-5-amine 50,

Page 8: Rashid et al. (2019) synthesized dione 54 in four steps, starting from 4-oxobutanehydrazide 53 (Scheme 8).

Page 8:  Bagul et al. (2023) synthesized a series of benzimidazole briged pyrazolo[1,5-a]pyrimidine 5760 by reaction between pyrazolo[1,5‑a]pyrimidine‑5‑carboxylate 55 and substituted benzene-1,2-diamines

Page 10: Chen et al. (2006) synthesized hybrid 63 in four steps from 4-chloro-2-(methylthio)pyrimidine 61 through the intermediate 2-(methylthio) pyrimidin-4-amine 62 (Scheme 10).

Page 11: Hunt et al. (2009) developed a family of benzimidazole-pyrimidine hybrids 6976,

Page 13: The intermediate 84 was halogenated with POCl3 and PCl3 at 110°C, yielding hybrid 85 (Scheme 11). The reaction of compound 85 with various aromatic amines led to hybrids 8692,

Page 13: Kunduru et al. (2014) synthesized benzimidazole-pyrimidine hybrids 94a94h by the reaction of substituted 3-phenylpropenones 93a93h with guanidine,

Page 14: AlNeyadi et al. (2017) reported synthesis of benzimidazole-pyrimidine acrylonitrile hybrids 98100 by reaction between 2-(1H-benzo[d]imidazol-2-yl)acetonitrile and 2-substituted pyrimidine-5-carbaldehyde in piperidine at 25°C in 81–89% yields.

Page 17: Khan et al. (2021) reported synthesis of hybrids 117124 by the reaction between 2-chloromethylbenzimidazole 82 and 6-substituted tetrahydropyrimidines 116 (Scheme 17).

Page 20: El Diwani et al. (2014) synthesized a series of new benzimidazole-pyrimidines 134136 by the reaction between tetrahydropyrimidine-5-carbonitriles 133a133c and 2-chloromethylbenzimidazole 82

Page 20: Selvam et al. (2010) synthesized benzimidazole-pyrimidine sulfonamide 140 (Fig. 17).

Page 23: Mathew et al. (2013) reported synthesis of 2-substituted benzimidazole-pyrimidine-2,4,6-triones 157162 by refluxing a mixture of enone 150155 and pyrimidine-trione 156 in acetic acid

Page 23: Farhan and Farooqui (2021} reported synthesis of benzimidazole-sulfinyl-pyrimidines 166174 by the reaction of 5-substituted benzimidazole-2-thiols 163a163c with 6-substituted 4-chloro-2-methyl pyrimidines 164a164i in basic medium of sodium hydroxide, and oxidation of intermediates

  1.  in line 173, "2-((1H-benzo[d]imidazol-2-yl)methylthio)-4-(substituted)-6-phenylpyrimidine-carbonitrile 18–37". Abdel-Mohsen and co-workers incorporated a thioether substrcture and a cyano group to obtain a new class of benzimidazole-pyrimidine hybrids 18–37. The thioether and the cyano group are their structural novelties. Similar problems are omnipresent in the manuscript.

Response: That is, from a structural point of view the compounds presented in this review are nothing special. This review aims to present benzimidazole-pyrazole hybrids with biological properties. These are the hybrid compounds found in the literature. We have reformulated:

Abdel-Mohsen and co-workers (2010) reported 2-(benzoimidazol-2-yl)methylthio)-6-phenylpyrimidine-carbonitriles 1837 (Fig. 2) of potent antitumor activity against 12 cell lines namely,

and in other places.....

  1. The manuscript can be published after the authors adress the above concerns.

Reviewer 3 Report

Comments and Suggestions for Authors

The review article summarizes the medicinal applications of an important class of medicinal chemistry that involves benzimidazole-pyrimidine hybrids. Authors have particularly focused on the last decade. It is an interesting piece of work, however, it lacks to address two main issues and would like the authors to revise their review accordingly. Authors should discuss current challenges and future perspectives. Another point that authors didn't discuss is the solubility, in particular, water solubility of these compounds as it is one of the basic factor in drug design. I would appreciate, if authors could please address the following points in revision.

  1. Please discuss that what are the challenges in drug design in general and in this class of hybrid molecules in particular? Please include future perspectives for further insights.
  2. Please discuss the solubility (particularly aqueous) of this class of compounds.
  3. Authors have added yields for some compounds. I would appreciate if it could be added for others as well (wheresoever possible) and discuss the role of substituents and reaction conditions to improve the synthetic yields.
  4. How do authors see the role of other chalcogens both within and ouside the ring?
  5. Figure 5 and Figure 13 (right image) needs better resolution as the text is not readable!
  6. The compound numbers should be bold. Please adopt uniformatiy.
  7. Abstract, line 14, "....to those of mono-heterocyclic compounds" should be, "....to those with single heterocyclic rings".
  8. Line 78, please remove dot after word red.
  9. Sometimes abbreviations are used first and at other instances abbreviations appear after their respective terms. This can be particularly seen in lines 94 to 100. Please adopt uniform style of representation.

Author Response

The review article summarizes the medicinal applications of an important class of medicinal chemistry that involves benzimidazole-pyrimidine hybrids. Authors have particularly focused on the last decade. It is an interesting piece of work, however, it lacks to address two main issues and would like the authors to revise their review accordingly. Authors should discuss current challenges and future perspectives. Another point that authors didn't discuss is the solubility, in particular, water solubility of these compounds as it is one of the basic factor in drug design. I would appreciate, if authors could please address the following points in revision.

  1. Please discuss that what are the challenges in drug design in general and in this class of hybrid molecules in particular? Please include future perspectives for further insights.

Response: Thank you for your observation. We have added a special subchapter referring to  "12. Current challenges and future prospects"

  1. Please discuss the solubility (particularly aqueous) of this class of compounds.

Response: Thank you for your observation. We have added a special subchapter referring to the solubility of benzimidazole-pyrimidine hybrids "11. Solubility of benzimidazole-pyrimidine hybrids"

  1. Authors have added yields for some compounds. I would appreciate if it could be added for others as well (wheresoever possible) and discuss the role of substituents and reaction conditions to improve the synthetic yields.

Response: Thank you for your observation. We have added wherever possible the yields of the reactions, as well as their correlation with the presence of substituents. Moreover, a paragraph in Chapter 12 refers only to this subject.

  1. How do authors see the role of other chalcogens both within and ouside the ring?

Response: The role of chalcogens both within and outside the ring could be beneficial in improving the therapeutic activity through the generated chalcogen-receptor interaction. Chalcogens can be easily bound or assimilated by certain receptor structures due to their similarity to important biochemical structures, such as enzymes.

  1. Figure 5 and Figure 13 (right image) needs better resolution as the text is not readable!

Response: Thank you for your observation. We have posted the figures with better resolution.

  1. The compound numbers should be bold. Please adopt uniformatiy.

Response: Thank you for your observation. We checked that all compounds are marked in bold.

  1. Abstract, line 14, "....to those of mono-heterocyclic compounds" should be, "....to those with single heterocyclic rings".

Response: Thank you for your observation. We wrote "....to those with single heterocyclic rings".

  1. Line 78, please remove dot after word red.

             Response: Thank you for your observation. This sentence was deleted at the suggestion  

               of a reviewer

  1. Sometimes abbreviations are used first and at other instances abbreviations appear after their respective terms. This can be particularly seen in lines 94 to 100. Please adopt uniform style of representation.

            Response: Thank you for your observation. We have corrected the mentioned place.

Round 2

Reviewer 3 Report

Comments and Suggestions for Authors

The authors have addressed majority of the concerns in best possible way. There are still couple of points that I would like the authors to kindly address before the review article is accepted for publication.

  1. The formatting issues still exist. For instance line 99, 179, 223, 330, 361,416, 449, 470, 538 and 687 seems to be incomplete due to formatting issues although these are continuous sentences.
  2. Scheme 1, there is no R2 for compounds 3a, 3b, 4a and 4b in the ChemDraws. The numbers 3-6 in caption should be bold.
  3. Some captions are still missing the "full stop".
  4.  

Author Response

The authors have addressed majority of the concerns in best possible way. There are still couple of points that I would like the authors to kindly address before the review article is accepted for publication.

  1. The formatting issues still exist. For instance line 99, 179, 223, 330, 361,416, 449, 470, 538 and 687 seems to be incomplete due to formatting issues although these are continuous sentences.

Response 1. Thank you for your comments. We have made changes in the text and in the schemes.

  1. Scheme 1, there is no R2 for compounds 3a, 3b, 4a and 4b in the ChemDraws. The numbers 3-6 in caption should be bold.

Response 2. Thank you for your comments. Yes, R2 had been added to the structure. We deleted R2 from the explanation. All compounds are marked in bold.

  1. Some captions are still missing the "full stop".

Response 3. Thank you for your observations. We added a period to all captions.
